# Influence of Adsorption Characteristics of Surfactants Sodium Dodecyl Sulfate and Aerosol–OT on Dynamic Process of Water-Based Lubrication

**Jingbo Fang, Pengpeng Bai, Chuke Ouyang, Chenxu Liu** [ID] **, Xiangli Wen, Yuanzhe Li** [ID] **, Yonggang Meng** [ID] **, Liran Ma * and Yu Tian ***

State Key Laboratory of Tribology, Department of Mechanical Engineering, Tsinghua University, Beijing 100084, China; fjb19@mails.tsinghua.edu.cn (J.F.); baipengpeng@mail.tsinghua.edu.cn (P.B.); oyck19@mails.tsinghua.edu.cn (C.O.); lcx@mail.tsinghua.edu.cn (C.L.); xlwen@tsinghua.edu.cn (X.W.); li-yz17@mails.tsinghua.edu.cn (Y.L.); mengyg@tsinghua.edu.cn (Y.M.)
* Correspondence: maliran@mail.tsinghua.edu.cn (L.M.); tianyu@mail.tsinghua.edu.cn (Y.T.)

**Abstract:** Surfactant solutions are widely used in industry, and their steady-state lubrication properties have been comprehensively explored, while the "dynamic process" between steady states attracts much less attention. In this study, the lubrication behaviors of sodium dodecyl sulfate (SDS) and sodium bis (2–ethylhexyl) sulfosuccinate (Aerosol–OT, AOT) solutions were comparatively and extensively discussed. Experimental results showed that the duration of the dynamic process of AOT solution lubrication was significantly shorter than that of SDS. The essence of the dynamic process was revealed from the aspects of the running-in of solid surfaces and the adsorption process of surfactant molecules. Unlike the general recognition that the friction force evolution mainly corresponds to the running-in of surfaces, this study indicated that the dynamic adsorption behavior of surfactant molecules mainly contributes to this process. Various experiments and analyses showed that the smaller steric hindrance and lower orientation speed of SDS molecules led to longer diffusion into the confined contact zone and a longer duration of friction force decrease. This work enhances our understanding of the dynamic friction process in water-based lubrication, which could also have important implications for oil-based lubrication and its industrial applications.

**Keywords:** aqueous lubricant; surfactant; dynamic process of friction; adsorption; orientation

## 1. Introduction

Studies have shown that approximately one-third of the total global primary energy consumption originates from tribological contacts, leading to significant industry demands to reduce friction and wear through proper lubrication [1,2]. Water-based lubricants are environmentally friendly and have a low cost and, thus, warrant studies on appropriate frictional additives [3–6]. In particular, amphiphilic surfactants are typically applied to improve boundary lubrication performance [1,6], and their lubrication properties and mechanisms require extensive research.

In previous studies, the effects of surfactants in steady-state lubrication (constant load and sliding velocity) have been widely recognized, wherein one end of the amphiphilic surfactant molecule is adsorbed onto a friction pair surface while the other end is exposed, resulting in the formation of a boundary film to reduce the direct contact of rough peaks [7–9]. In steady-state lubrication, surfactant molecules experience a dynamic equilibrium of adsorption, desorption, and mechanical disturbance on the solid surface [10,11]. The anti-friction effect is generally determined by the surface coverage of the adsorption film, which can be influenced by the adsorption strength and the packing density of the surfactants at the interface [12–15].

Research on the process before the friction reaches a steady state mainly focuses on the running-in. In the running-in process, two solid surfaces that contact with rough peaks

undergo significant elastic and plastic deformations and generate obvious wear until the wear rate or coefficient of friction (COF) reaches a steady state [16]. The essence of this process includes the physical and chemical evolution of the surface morphology, surficial material composition and structure, and third-body distribution between the friction pair, etc. [17,18]. The subsequent lubrication performance after running-in can be greatly affected by the working conditions of the running-in [16,17,19].

Unlike the running-in process, there are relatively few studies on the "dynamic process"; that is, on the surface that has finished the general running-in, the COF changes due to the change in working conditions (load or sliding velocity), and then it gradually reaches a stable value due to the molecular behavior of the lubricant. Some studies have noted that, for some organic friction modifiers, the COF in the first friction cycle may be higher than that of subsequent cycles, because friction stimulates or accelerates the formation of adsorption films [20–23]. Another typical case is the friction-induced extension and orientation of long-chain molecules and liquid crystals, causing the reduction of interfacial viscosity and molecular internal friction. Accordingly, COF is gradually decreased [24–29].

Sodium dodecyl sulfate (SDS) is one of the commonly used additives in water-based lubricants. As shown in Figure 1a, the SDS molecule is single-chained, with a critical micelle concentration (CMC) of 8 mM [30]. Zhang studied the relationship between the adsorption structure and COF [31]. Siqing and Liu applied different electrode potentials to the friction pair to control the adsorption/desorption of SDS and its tribological properties [32–34]. As another widely studied surfactant in aggregation and wetting, sodium bis (2–ethylhexyl) sulfosuccinate (Aerosol–OT, AOT) is double-chained, with the same number of carbon atoms in the main carbon chains as SDS, as shown in Figure 1b. AOT is more hydrophobic, with a CMC of only 2.5 mM [35], resulting in a strong aggregation and transport ability which means that it can sequentially form micelles, vesicles, and several phase structures as the concentration increases [35–38]. Furthermore, AOT vesicles exhibit a higher diffusion rate and adsorption strength to hydrophobic surfaces than SDS micelles [38–40]. However, AOT has not been investigated as a lubricant additive, probably due to the large-steric-hindrance molecular structure that makes it hard to form a dense adsorption layer. Therefore, comparative studies on the lubrication behaviors of SDS and AOT solutions could be carried out to explore the effects of molecular structure.

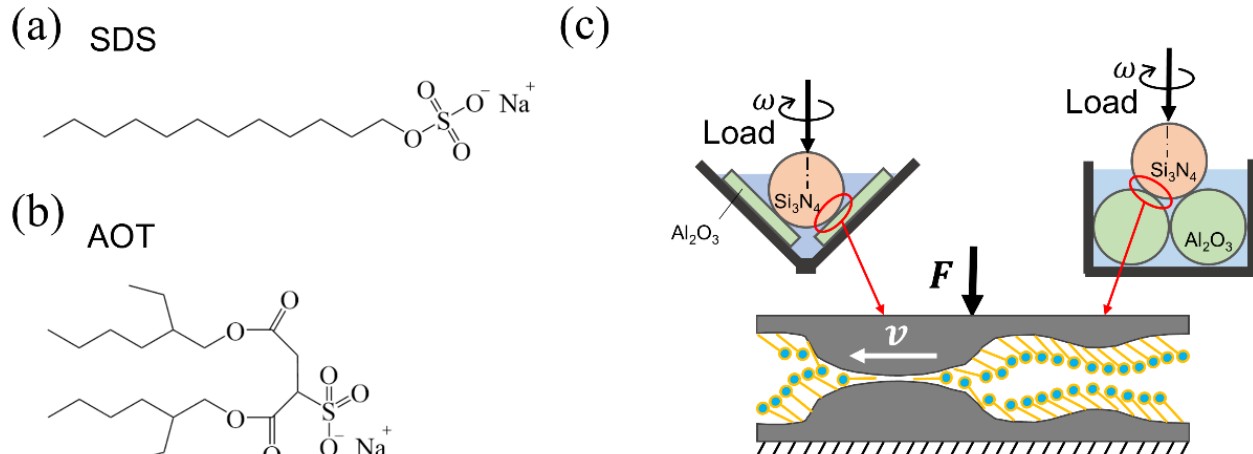

**Figure 1.** Schematic of the two molecules and the friction test apparatuses used in this study. (**a,b**) Structure diagram of sodium dodecyl sulfate (SDS) and sodium bis (2–ethylhexyl) sulfosuccinate (Aerosol–OT, AOT). (**c**) Illustration of the ball-on-three-plate and four-ball friction test apparatuses and contact region. F and v denote the load and velocity applied to the contact zone, respectively. For simplicity, only two plates and two lower balls are shown in the figure.

In summary, previous studies mainly focus on the steady-state lubrication situation and the traditional running-in process while lacking an exploration of the dynamic process

during which the COF decreases owing to the molecular behavior of lubricants. In this study, SDS and AOT aqueous solutions were used as lubricants, and two kinds of commonly used ceramics, silicon nitride and aluminum oxide, were selected as friction pairs, due to their high corrosion and wear resistance [41]. The dynamic lubrication processes of the two surfactants were comparatively studied under various working conditions, and the duration of the dynamic process was the main focus. The essence of the dynamic process was revealed from the evolution of the solid surface topography and the adsorption behaviors of the surfactant molecules. The adsorption characteristics were compared from the aspects of adsorption amount, surface coverage, and diffusion and orientation speed, and their effects on the dynamic process of lubrication were determined.

## 2. Materials and Methods

### 2.1. Materials

The two surfactants used in the study were SDS (purity > 99%; Shanghai Macklin Biochemical Technology Co., Ltd., Shanghai, China) and AOT (purity > 99%; Tianjin Shine-sopod Technology Co., Ltd., Tianjin, China). Their aqueous solutions with concentrations of 1.6–30 mM were prepared with deionized water (DW) at room temperature one day before the friction tests were carried out, and the measured CMCs of SDS and AOT were 7.3 mM and 2.5 mM, respectively. In addition, aluminum oxide plates ($Al_2O_3$; 15 mm × 6 mm; Ra ~ 2.7 nm; Xi'an Laikete Electronic Technology Co., Ltd., Xi'an, China), silicon nitride balls ($Si_3N_4$; φ 12.7 mm; Ra ~ 2.5 nm; Shanghai Nuansheng Electronic Commerce Co., Ltd., Shanghai, China), and aluminum oxide balls ($Al_2O_3$; φ 12.7 mm; Ra ~ 30 nm; Dongguan Bluewhale Ceramic Technology Co., Ltd., Dongguan, China) were selected as the friction pairs. They were cleaned in an ultrasonic bath successively with petroleum ether, ethanol, and DW for 5 min each and dried with compressed air.

### 2.2. Tribology Tests

Friction tests were performed with two apparatuses (Figure 1c). Firstly, a ball-on-three-plate friction test apparatus (MCR301, Anton Paar Trading Co., Ltd., Graz, Austria) was mainly used, and the friction pairs were three $Al_2O_3$ plates and one $Si_3N_4$ ball. Secondly, the combined experiments of friction and wear scar observation were performed on a four-ball test apparatus and its supporting microscope (MS–10JS, Xiamen Tenkey Automation Co., Ltd., Xiamen, China); the friction pairs were three $Al_2O_3$ balls and one $Si_3N_4$ ball. The magnification of the microscope was 300×. All friction tests were carried out at 25 °C with DW or surfactant solutions as lubricants. The hydrolysis and decomposition of AOT and SDS can be neglected in this work. For each newly prepared friction pair, DW was first used as the lubricant for running-in for a maximum of 1500 s until the changes in COF over 120 s were within 0.03 (on the ball-on-three-plate tester) or 0.08 (on the four-ball tester). After running-in, the friction pairs were rinsed four times, alternately with DW and ethanol, and were cleaned with Kimwipes tissue. Then, experiments with surfactant solutions as lubricants could be started. In each friction experiment, the load used for running-in was the maximum value of the loads in the subsequent friction tests in order to establish a stable contact state before the friction tests with the SDS or AOT solutions and so that the wear scars would not expand significantly during friction. The applied normal force and speed were different in different tests and are further explained in the subsequent sections.

### 2.3. Adsorption Measurement at Solid–Liquid Interface

The adsorption mass of SDS or AOT molecules at different concentrations was measured using a quartz crystal microbalance (QCM; Q–sense E4 system, Biolin Scientific, Sweden) with an $Al_2O_3$-coated quartz crystal sensor at 25 °C, and the flow rate of the liquids was 100 μL/min. Moreover, the adsorbed film was regarded as rigid [42,43], and the adsorbed mass (Δ*m*) was calculated using the Sauerbrey equation [43] presented in Equation (1), when changes in the resonance frequency (Δf) and dissipation (ΔD) of the sensor conformed to ΔD < 10Δf:

$$\Delta m = -C \frac{\Delta f}{n}, \tag{1}$$

where $C$ is a constant with a value of 17.7 ng $Hz^{-1}$ $cm^{-2}$ that is related to the properties of the quartz crystal, and n denotes the overtone of the oscillations.

## 3. Results

On the friction pair that had been running in with DW under a load of 20 N, a step-up load was applied as $F_N$ = 1, 2, 5, and 10 N, respectively, and the friction test lasted 40 s at each load. The angular velocity ($\omega$) of the $Si_3N_4$ ball was 300 rpm. Different concentrations of solutions were used as lubricants, and the results are shown in Figure 2. Since the CMC of SDS (7.3 mM) and AOT (2.5 mM) are different, the lubrication behavior of the two solutions was compared at concentrations lower than the CMC (Figure 2a), higher than the CMC (Figure 2b), about two times the CMC (Figure 2c), and about four times the CMC (Figure 2d). After each load, the COFs of SDS and AOT increased immediately and then experienced a "dynamic process" during which the COF gradually decreased until finally approaching a stable value. The decreasing speed of the COF of AOT was higher, and the duration of the dynamic process of AOT was significantly shorter than that of SDS at all concentrations, while the stable COF values of SDS were lower.

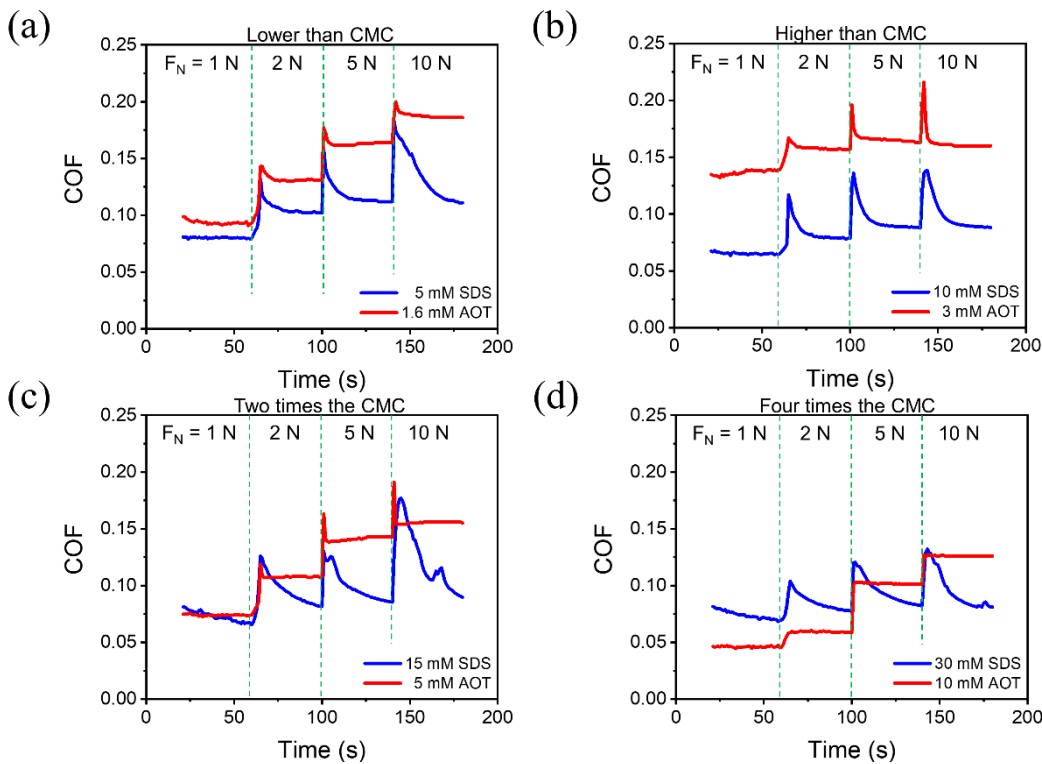

**Figure 2.** COF of the tests at 300 rpm (the sliding velocity was 0.141 m/s) and step-up load. The load was applied as $F_N$ = 1, 2, 5, and 10 N, respectively. Different concentrations of SDS and AOT solutions were used as lubricants, including the concentrations that were (**a**) lower than the CMC, (**b**) higher than the CMC, (**c**) about two times the CMC, and (**d**) about four times the CMC.

On the friction pairs that had been running in with DW under a load of 1 N, friction tests were carried out with a constant load of 1 N and different angular velocity, as shown in Figure 3. Different concentrations of SDS and AOT, lower than or about twice the CMC, were used as lubricants. Figure 3a,c,e shows the curves of COF over time under angular velocities of 60, 150, and 600 rpm, respectively. For a more intuitive comparison of the time required for the dynamic process, a normalized COF was obtained by dividing the COF by its stable value, shown in Figure 3b,d,f. The stable COF value was taken as the average

value of the last 10 s of each experiment; thus, the normalized COF would eventually tend to 1. The gray lines in the figures represent the ±5% of the stable COF, between which the COF was considered stable. Furthermore, they enabled an intuitive comparison of the time required for the dynamic process. Similar conclusions to the above figures could be seen such that, under all system angular velocities, the COF of all concentrations of SDS experienced a slower and longer decrease until lower stable COF values were reached. The duration for 15 mM SDS was longer than that for 5 mM SDS. Moreover, the lower the velocity, the longer was the dynamic process.

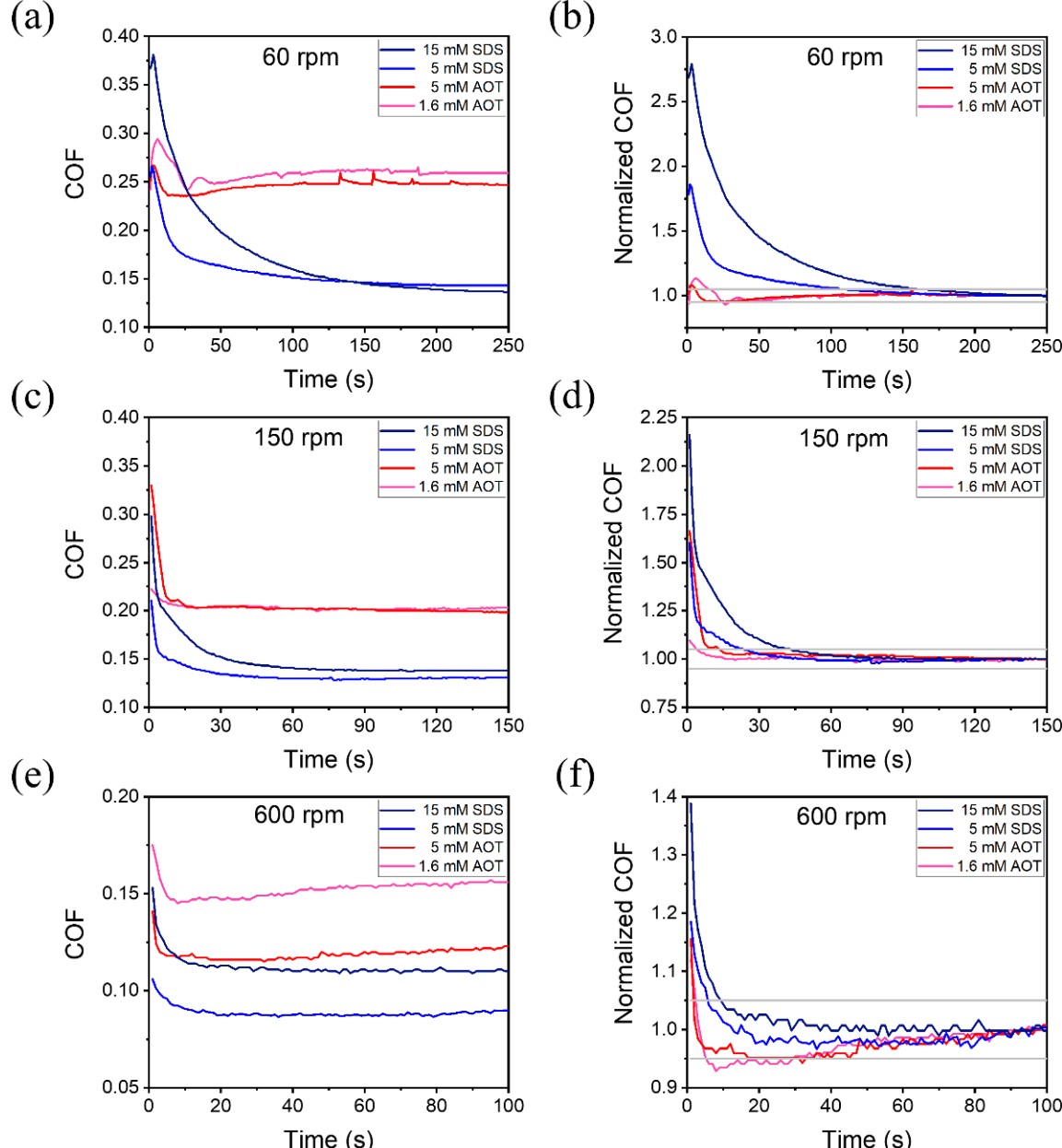

**Figure 3.** COF of the tests with a constant load of 1 N and different angular velocities of (**a,b**) 60 rpm (0.0282 m/s), (**c,d**) 150 rpm (0.0705 m/s), and (**e,f**) 600 rpm (0.282 m/s). Normalized COF values (**b,d,f**) were obtained by dividing the COF values by their respective stable values.

The Stribeck curves of 5 mM SDS and AOT were obtained in continuous variable speed experiments to characterize the lubrication state under various conditions, as shown in Figure 4. The applied load of the running-in and the subsequent friction tests was 2 N;

$\omega$ varied from 0.1 to 3000 rpm. The friction lasted 3 s at each $\omega$ to avoid large differences in solid surface, and the COF was an average of 3 s. For both SDS and AOT, the COF in round 1 was higher than those in the other rounds. This was because the $Si_3N_4$ ball was lifted while changing lubricants (from DW to the surfactant solution) and was then placed at the measurement position again. Accordingly, the surfaces experienced a short running-in at the beginning of each experiment. The friction curves of SDS in rounds 2–4 did not coincide well, and the COF in the boundary lubrication region decreased significantly among the rounds until the COFs of rounds 5–8 were almost coincident. In contrast, the friction curves of AOT in rounds 2–8 all had a high coincidence degree. Thus, the dynamic process duration of SDS (rounds 1–4) was much longer than that of AOT (round 1). Moreover, in the dynamic process, COF varies more with speed in the boundary lubrication region than in the stable phase. Furthermore, the lubrication states in Figures 2 and 3 were mixed lubrication, according to the stable Stribeck curves.

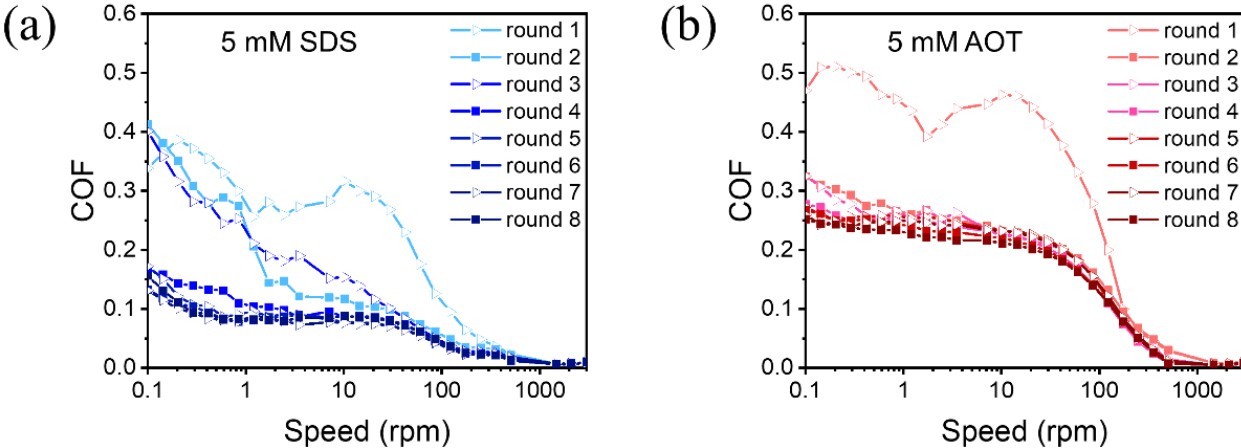

**Figure 4.** COF evolution in the procedures of the continuous variable speed experiment with (**a**) 5 mM SDS and (**b**) 5 mM AOT as lubricants. $F_N$ = 2 N. The value of $\omega$ increased from 0.1 to 3000 rpm in round 1, reduced from 3000 to 0.1 rpm in round 2, increased again in round 3, and so on until it decreased to 0.1 rpm in round 8. Each round was continuous without stopping or changing the direction of rotation.

In summary, friction experiments were conducted under various working conditions, including combinations of several speeds and loads, using different concentrations of SDS and AOT aqueous solutions as lubricants. It was found that in the dynamic process before the system reached a steady state, the COF of AOT decreased faster than that of SDS, and the duration of the dynamic process of AOT was shorter than that of SDS.

## 4. Discussion

An alternate loading–unloading experiment was performed to understand the essence of the dynamic process, as shown in Figure 5. The friction pair had been running in under 10 N. When the load was increased from 5 N to 7 N for the first time, the COF increased abruptly, experienced a dynamic process, and finally stabilized. When unloaded to 5 N and loaded to 7 N for the second time, the COF increased and remained stable, with no spikes generated. Thus, it could be inferred that there were two possible reasons why the COF decreased with time before the system reached a steady state. Firstly, the contact zone experienced a slight running-in. Secondly, on the fresh surface generated by running-in, the arrangement properties of the surfactant molecules, including the number of molecules adsorbed in the contact zone and the molecular conformation and orientation, would gradually reach a dynamic equilibrium such that the COF would reach a stable value.

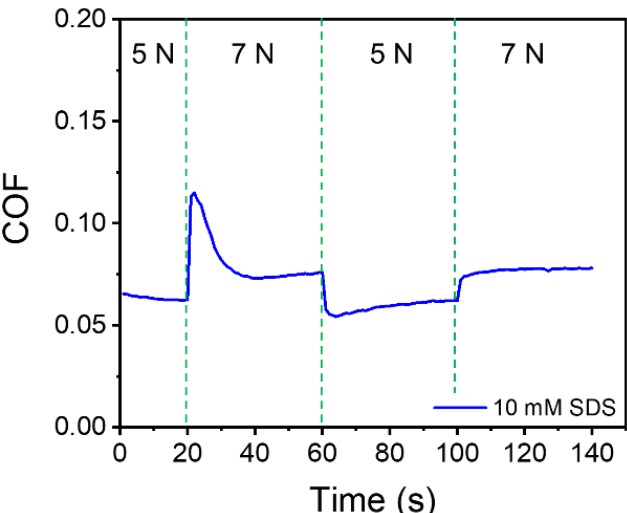

**Figure 5.** COF in the test of alternate loading–unloading. The load was applied as $F_N$ = 5, 7, 5, and 7 N, respectively, and $\omega$ = 300 rpm; 10 mM SDS was used as a lubricant.

To characterize the effect of running-in, the four-ball friction test apparatus and its supporting microscope were used to conduct friction experiments and observe wear scars, and the results for 5 mM SDS and AOT are, respectively, shown in Figures 6 and 7. The experiment consisted of two stages. Stage 1 contained the rising edge of the COF and was stopped after the COF reached its peak value, while stage 2 contained the falling edge of the COF and was stopped when the COF tended to be stable. The experiment was designed to compare the wear scars at different moments of the dynamic process in order to verify the effect of running-in on the dynamic process.

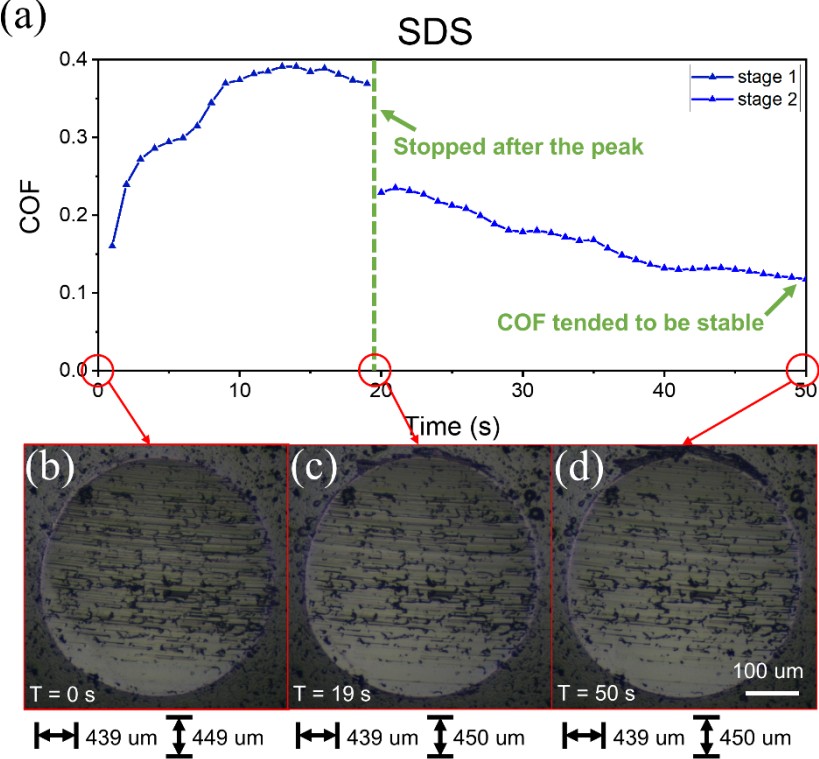

**Figure 6.** Friction curves with 5 mM SDS solution lubrication and wear scar at each stage. (**a**) Two stages of COF curves over time. Before stage 1, the friction pair had been running in with DW, and the wear scar was as shown in (**b**). (**c**,**d**) The wear scar after stage 1 and stage 2, respectively.

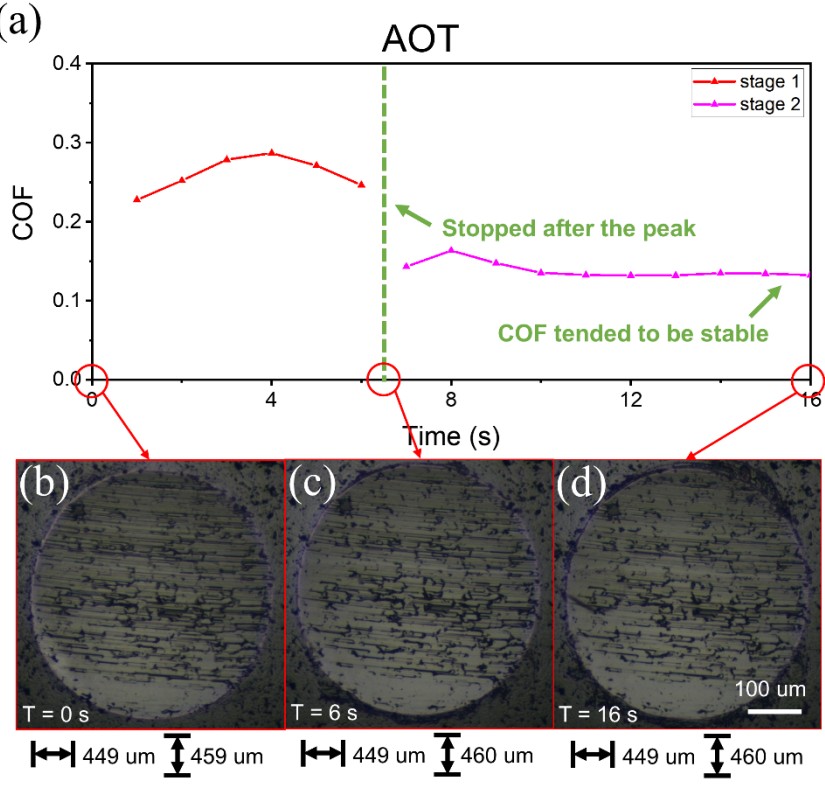

**Figure 7.** Friction curves with 5 mM AOT solution lubrication and wear scars at each stage. (**a**) Two stages of COF curves over time. The load and speed were the same as those of SDS. Before stage 1, the friction pair had been running in with DW, and the wear scar was as shown in (**b**). (**c**,**d**) The wear scar after stage 1 and stage 2, respectively.

Figure 6a shows the COF over time with 5 mM SDS as a lubricant. The applied load was 5 N, and the angular velocity of the upper ball was 300 rpm. Before starting, the friction pair had been running in with DW under a load of 5 N; the wear scar of one of the lower balls is shown in Figure 6b. Then, the SDS solution was added as a lubricant to start the friction test; this corresponds to stage 1 in Figure 6a. During this period, the COF first increased and then began to decrease. Thus, the friction was stopped, and the solution was poured out and blow-dried to observe the wear scar of the lower balls, as shown in Figure 6c. After that, new SDS solution was added to start the friction test again; this corresponds to stage 2 in Figure 6a. In this stage, the COF increased slightly again and gradually decreased in the following 30 s until the COF tended to be stable. Thus, the friction test was stopped again to enable observation of the wear scar, as shown in Figure 6d. In comparing the size and internal morphology of wear scars in Figure 6b–d, the size in Figure 6c along the friction direction was only slightly larger than that in Figure 6b, and the internal grooves were slightly shallower, while the size and grooves of Figure 6d showed no obvious changes compared with Figure 6c. These results demonstrate that a slight running-in occurred in the contact zone during stage 1, while the change of wear scar was not obvious in stage 2, and so the continuous decline of the COF in stage 2 was not only related to running-in. In addition, it could be seen from the figures that no obvious corrosion had occurred on the ball.

Similar experiments were also performed with 5 mM AOT solution (Figure 7), and a similar phenomenon was observed. The COF first increased for several seconds and then decreased during each stage. The size and internal grooves of the wear scars changed slightly, similar to Figure 6. The main difference with Figure 6 was the duration of the falling edge of the COF. Thus, the main difference in the duration of the dynamic process between AOT and SDS should be related to molecular behavior rather than running-in.

Since the anti-friction mechanism of surfactant molecules was the adsorption on the solid surface and the avoidance of direct contact of solids, the adsorption characteristics of SDS and AOT solutions were compared on QCM, as shown in Figure 8. Typical curves of the adsorbed number of molecules per unit area are depicted in Figure 8a, where the adsorption amount gradually increased and tended to stabilize. The statistical results of the final adsorption amount of different concentrations of SDS were larger than that of the AOT solutions at all concentrations (Figure 8b), and the dissipations were all less than $2.0 \times 10^{-6}$. The result was reasonable, because the molecular structure of SDS is a single straight chain, while that of AOT is a double-branched chain, such that SDS can be arranged more densely than AOT on the $Al_2O_3$ sensor of QCM [7,44]. The peak value in the adsorbed amount curve of 15 mM SDS (Figure 8a) was due to the fact that SDS reached the surface of the $Al_2O_3$ sensor in the form of micelles and then underwent a transition of arrangement structure upon adsorption, and the difference in the arrangement structures of different concentrations of SDS solutions on the surface was probably the reason for their different adsorption amount. The adsorption onto the QCM sensor is a thermodynamic equilibrium in a free space, which may be different from that in the confined space between the friction pairs, but the results from QCM could be used for reference.

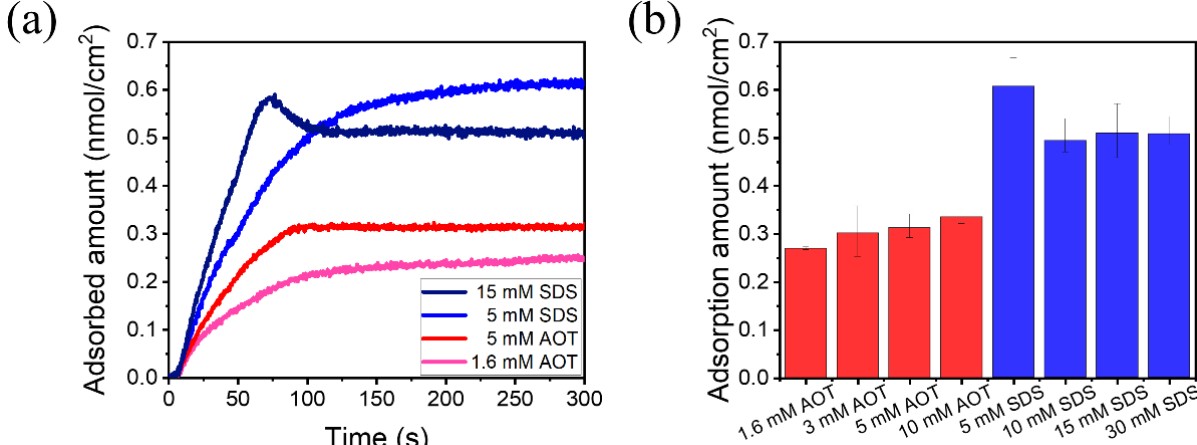

**Figure 8.** Adsorption of SDS and AOT solutions measured using QCM. (**a**) Typical scatterplot of the adsorbed molecular number per unit area varying with the introduction time of the liquid. (**b**) Statistical results of adsorption amount of different concentrations of SDS and AOT solutions.

The difference in the molecular structure and adsorption amount of SDS and AOT led to different lubrication effects, as shown in the above figures. At the end of the dynamic process, the COF reached a stable value, and the stable COF value of SDS as lubricant was lower than that of AOT in boundary and mixed lubrication regions. There are two possible reasons. Firstly, the boundary film of SDS was denser (Figure 8), which meant that it could achieve a better anti-friction effect [44]. Secondly, some parts of the contact zones were very limited in space, with only one or a few molecular diameters. Compared to AOT, SDS molecules were more likely to enter these contact zones due to their smaller steric hindrance in molecular structure (Figure 9); thus, a higher surface coverage could be achieved by SDS.

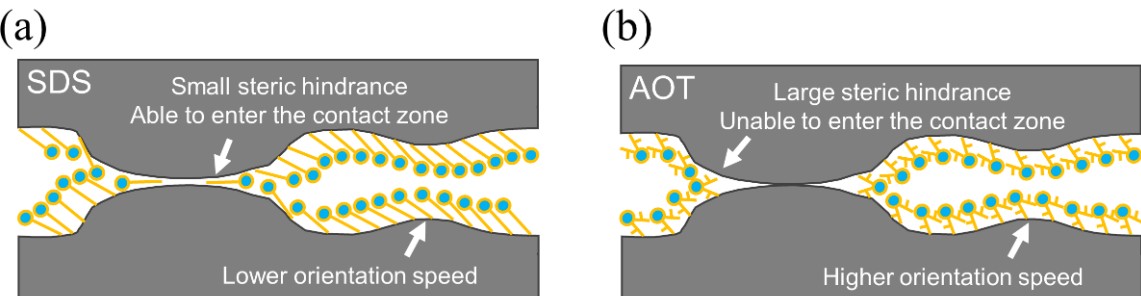

**Figure 9.** Mechanism illustration for the molecular behavior in the dynamic process. (**a**) SDS had a smaller steric hindrance such that it could enter the contact zones with confined space slowly and continuously. SDS molecules had a lower orientation speed. (**b**) AOT had a larger steric hindrance, making it difficult to enter the same contact zones as SDS. AOT molecules had a higher orientation speed.

Since the diffusion time of confined molecules could be considerably longer than that of bulk molecules [45], the above-mentioned process of SDS molecules entering confined contact zones would take longer to reach equilibrium. This could be one of the possible reasons why the duration of the dynamic process of SDS was much longer than that of AOT. Moreover, the adsorbed molecules were subjected to the shear forces of the solid and fluid during the friction process, causing the chains to gradually extend and orient to a more uniform tilt angle and direction [25–27]. The orientation of the molecules can promote molecular alignment, reduce the molecular internal friction [28,29], and may also increase the packing density [46]. Thus, the COF decreased with the friction process. The larger packing density and greater intermolecular dispersion force of SDS made it harder to orient than AOT (Figure 9) [25,47]. Thus, SDS needed longer to reach a stable COF, and the COF of AOT decreased more rapidly than that of SDS at the beginning of the dynamic process.

All friction tests in this work were carried out at 25 °C, but, in practical application, the ambient temperature could be higher. Temperature can affect molecular behaviors in dynamic processes such as adsorption and diffusion. On the one hand, for the slight running-in stage at the beginning of the dynamic process, the adsorption amount [48] and anti-friction effect of surfactant molecules decrease with the increase in ambient temperature. Therefore, the running-in stage may last longer. On the other hand, the higher the temperature, the greater is the molecular diffusion rate into the confined contact zone. Therefore, the COF may decrease more rapidly over time, and the duration of COF decrease may be shorter. The characteristics of the dynamic process under different temperatures are determined by these factors together.

## 5. Conclusions

In this study, the influence of the adsorption characteristics of surfactants SDS and AOT as lubricant additives on the dynamic process of the lubrication process was comparatively studied. It was found that the duration of the dynamic process of AOT was significantly shorter than that of SDS, while SDS could achieve a lower stable COF value. The essence of the dynamic process was the slight running-in of the solid surface and the dynamic adsorption behaviors of surfactant molecules, of which the latter had a greater impact on the differences in the dynamic processes of the two surfactants. The longer duration of the dynamic process of SDS was attributed to its slow access to a more confined contact zone, which achieved a higher surface coverage and a lower stable COF value. The faster COF decrease of AOT at the beginning of the dynamic process was attributed to its higher orientation speed. This study focused on the unsteady friction process that is often overlooked in research and investigated the molecular behavior of lubricants during this process. The results provide an in-depth understanding of unsteady lubrication behavior

in water-based lubrication, which could also provide important guidance for oil-based lubrication and its industrial applications.

**Author Contributions:** Conceptualization, Y.T.; investigation, J.F. and C.O.; methodology, J.F., C.O. and L.M.; resources, Y.M., L.M. and Y.T.; validation, P.B., C.L., Y.M. and Y.T.; writing—original draft, J.F.; writing—review and editing, J.F., C.O., C.L., X.W., Y.L. and Y.T. All authors have read and agreed to the published version of the manuscript.

**Funding:** This research was funded by the National Nature Science Foundation of China (51425502) and the Ten Thousand People Leading Plan Innovation Leading Talents of China (No. 20191700617).

**Data Availability Statement:** Not applicable.

**Conflicts of Interest:** The authors declare no conflict of interest.

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
