# Peer review of "Influence of Adsorption Characteristics of Surfactants Sodium Dodecyl Sulfate and Aerosol–OT on Dynamic Process of Water-Based Lubrication"

_lubricants, doi:10.3390/lubricants10070147_

Round 1
Reviewer 1 Report
1. How do the authors consider corrosion and friction corrosion in water-based lubrication systems ?
2. What is the basis for the selection of the friction pair in this work?
3. For sample AOT, there will be no hydrolysis or decomposition problems during friction in water-based systems ? How did the author think about it ?
Reviewer 2 Report
All friction tests were carried out at 25 ºC, but in practical application, the friction temperature is higher, can the authors predict or discuss the dynamic process difference compared with that of the 25 ºC?
Round 2
Reviewer 1 Report
The modification is reasonable and recommended to receive.